# Energy Dense Salty Food Consumption Frequency Is Associated with Diastolic Hypertension in Spanish Children

**DOI:** 10.3390/nu12041027

**Published:** 2020-04-09

**Authors:** Gloria Pérez-Gimeno, Azahara I. Rupérez, Rocío Vázquez-Cobela, Gonzalo Herráiz-Gastesi, Mercedes Gil-Campos, Concepción M. Aguilera, Luis A. Moreno, María Rosaura Leis Trabazo, Gloria Bueno-Lozano

**Affiliations:** 1GENUD Research group, Universidad de Zaragoza, Instituto Agroalimentario de Aragón (IA2), Instituto de Investigación Sanitaria (IIS) Aragón, 50009 Zaragoza, Spain; gloriaperez@unizar.es (G.P.-G.); airuperez@unizar.es (A.I.R.);; 2Investigation Unit in Nutrition, Growth and Human Development of Galicia GI Pediatric Nutrition-Santiago Health Research Institute (IDIS), Pediatrics Department, Universitary Clinical Hospital of Santiago, Santiago de Compostela University, 15706 Santiago de Compostela, Spain; 3Unidad de Endocrinología Pediátrica, Hospital Clínico Lozano Blesa, Facultad de Medicina, Universidad de Zaragoza, 50009 Zaragoza, Spain; 4Metabolic Pediatric and Investigation Unit, Hospital Universitario Reina Sofía, 14004 Córdoba, Spain; 5Centro de Investigación Biomédica en Red de Fisiopatología de la Obesidad y Nutrición (CIBERObn), Instituto de Salud Carlos III, 28029 Madrid, Spain; 6Departamento de Bioquímica y Biología Molecular II, Instituto de Nutrición y Tecnología de los Alimentos, Centro de Investigación Biomédica, Universidad de Granada, 18071 Granada, Spain

**Keywords:** consumption frequency, dietary approach stop hypertension, energy-dense salty food, hypertension, pubertal stage, sugar-sweetened beverages

## Abstract

High blood pressure (BP) is a risk factor for cardiovascular disease and sodium consumption is related to high BP. Moreover, sugar-sweetened beverages (SSB) and the Dietary Approach to Stop Hypertension (DASH) influence BP. For this reason, we investigated whether: 1) children with risk of elevated BP had a higher consumption frequency (CF) of energy-dense salty foods (EDSF), high-sugary foods (HSF) and SSB or a low DASH score; and 2) children with a higher CF of EDSF showed a worse anthropometric and metabolic profile. Anthropometry, BP and general biochemical parameters were measured in 687 Spanish children (5–16 years) with normal or excess weight. A food frequency questionnaire was used to calculate EDSF, HSF and SSB consumption, and modified DASH score. Results showed that sex and pubertal stage influenced modified DASH score. Diastolic hypertension was associated to higher CF of EDSF in the whole sample and to higher CF of SSB in pubertal children, both independently of nutritional status. In addition, CF of EDSF was positively associated with CF of HSF and SSB and inversely associated with modified DASH score. Targeted policies and intervention programs, specific for different age ranges, should be established that aim to reduce salt consumption from snacks and processed foods, which could reduce HSF and SSB consumption as well.

## 1. Introduction

According to the World Health Organization (WHO), obesity prevalence has nearly tripled between 1975 and 2016, with childhood obesity being considered an important public health problem worldwide [1].

Childhood obesity is closely associated to the presence of cardiovascular and metabolic alterations such as hypertension. This is supported by the parallel increase in the prevalence of elevated blood pressure (BP) in childhood [2]. Many studies show that high BP starting during childhood has a high probability of continuing into adulthood, which involves an increased cardiovascular risk for hypertensive children along their life [3,4,5]. 

Dietary habits involving a high salt or sugar intake have been shown to influence the development of obesity and hypertension in children and adolescents. Excessive salt intake is positively associated with BP, having being proposed as the main preventable factor of hypertension [6]. However, not all sodium consumption is due to table salt intake. Given the generalized presence of added salt in processed foods, these are the major contributors of daily sodium intake [7]. Studies in children show a decrease in both, systolic blood pressure (SBP) and diastolic blood pressure (DBP), in parallel to the reduction in salt consumption [8,9,10,11]. Yang et al., found in children from USA that salt consumption was higher in normal-weight children than in overweight or obesity children [12]. Regarding sugars, some authors have showed higher BP levels in people who had a higher sugar intake, independently of obesity [13]. Specifically, sugar-sweetened beverages (SSB) are the sugar source that shows an association with higher BP in people over 12 [14]. The association between sugar and BP has also been observed in children [15,16,17]. In addition, some studies showed that salt intake is the major determinant of fluid and SSB consumption during childhood [9]. 

Given the association between food intake and BP, dietary recommendations such as the Dietary Approach to Stop Hypertension (DASH) have been described as potential preventive strategies against high BP and cardiovascular disease in children and adults [18,19,20,21,22]. The DASH diet consists on a high consumption of fruits, vegetables, legumes, nuts, whole grains, low-fat dairy products, fish, chicken and a low consumption of red meat and SSBs [18,19]. Studies have shown a negative association between DASH and arterial hypertension in adults [20] and children [19,21,23].

In order to retrieve unbiased and valid results, other aspects such as puberty and sex shall also be taken into account. Indeed, puberty is a period with many changes, with one of these being food intake, which increases through the pubertal development. In addition, there are sex differences in food and energy intake. In males, the significant changes occur during late puberty, whereas in females changes occur during early and mid-puberty [24], both related to the period of most rapid growth [25]. 

With all this in mind, the aim of this study was to investigate the association between elevated BP and the consumption frequency (CF) of: energy-dense salty foods (EDSF), high-sugary foods (HSF) and SSB and the DASH score, taking into account pubertal stage and sex. In addition, to investigate anthropometric and cardiometabolic profile in children with a higher consumption of EDSF.

## 2. Materials and Methods

### 2.1. Study Sample

A total of 687 children between 5 and 16 years old participated in the GENOBOX (11/01425, PI11/02042, PI11/02059) case control study, which was carried out in three Spanish cities: Santiago de Compostela, Zaragoza and Córdoba. All participants and their families were informed about the purpose of the study before giving their written consent. Inclusion criteria were: children between 5 and 16 years with normal-weight (control group) or overweight/obesity (case group). Exclusion criteria were: presence of chronic or inflammatory disease, congenital disease or psychomotor disability and use of any medication that alters blood pressure, hormonal, glucose or lipid metabolism, having exercised intensely in the 24 hours previous to the examination or having participated in a research study in the previous three months. The case group was recruited randomly to the children attending the hospital for diagnosis of minor gastrointestinal disorders; that were not confirmed after clinical and laboratory investigations or suspecting overweight or obesity. Whereas the control group was randomly recruited after they had come to the emergency department due to an acute banal infectious pathology or for diagnosis of minor gastrointestinal disorders; that were not confirmed after clinical and laboratory investigations. The study was implemented following the Declaration of Helsinki recommendations and with the approbation of the Ethics Committees of each participating center.

### 2.2. Clinical Examination

Pubertal stage was determined according to Tanner´s criteria by a pediatrician [26]. Children in Tanner Stage I were considered prepubertal, and children with Stages II–V were considered pubertal.

### 2.3. Anthropometric Measurements

Weight and height were measured by trained researchers using standardized procedures. Weight was recorded with the children in their underwear and without shoes using an electronic scale. Height was obtained also barefoot using a stadiometer. Body mass index (BMI) was calculated as body weight in kg divided by the square of height in meters. Children were classified as having normal-weight, overweight or obesity using the Cole et al., sex and age specific cut-offs for children equivalent to adult values of 25 kg/m2 and 30 kg/m2 [27]. 

### 2.4. Blood Pressure

SBP and DBP were measured twice with an electronic manometer, (Omrom, M6 AC) with a 5-minute interval and the mean was calculated for analyses. If measures differed more than 20%, an additional measurement was taken and the average was calculated with the two most similar values. In addition, mean arterial blood pressure (MAP) was calculated as ((2*DBP) + SBP))/3 [28]. Children were classified as at risk or without risk of having elevated BP or hypertension according to their SBP and DBP values using the 2017 Clinical Practice Guideline for Screening and Management of High Blood Pressure in Children and Adolescents [29]. SBP or DBP values equal or above the sex-, age- and height-specific (P90th) were indicative of elevated BP and those above the P95th were indicative of hypertension (HTN) (Stage I and II were considered together in one category).

### 2.5. Dietary Consumption

Dietary intake was assessed using an adaptation of a previously validated food frequency questionnaire (FFQ) [30]. It had 13 food categories with a total of 83 food items. The FFQ was filled in by a trained dietician-nutritionist who interviewed the main caregiver or child or adolescent, if they were older and had better idea of their own diet than their caregiver, in relation to the child´s habitual intake in the previous four weeks. There were nine possible answers of consumption frequency for each food item: “never or almost never”, “1–3 times per month” “once per week”, “2–4 times per week”, “5–6 times per week”, “Once per day”, “2–3 times per day”, “4–6 times per day”, “6 or more times per day” and “I have no idea”. Finally, these categories were transformed into a continuous “times per week” variable, with possible values between 0 and 42. 

Food groups were defined according to their content on BP-related nutrients (i.e., salt, sugar): EDSF that included foods with average sodium content above 700 mg / 100 gr: cheese, pizza, chips, sausages, butter, sauces and precooked foods; HSF such as sugar, honey, quince, jam, cocoa powder, cocoa cream, sweet cereal, cakes, buns, pastry, packaged cakes, donuts, chocolate, biscuits, candy and ºice-cream; and SSB including sweetened carbonated beverages, sweetened noncarbonated beverages and packaged fruit juices. Children were categorized into three CF groups depending on their percentile CF of EDSF (<P25th, P25th–75th, >P75th).

A modified DASH (MDASH) score was calculated using the following components calculated from FFQ data: fruits, vegetables, nuts and legumes, whole grain, low-fat dairy products, red and processed meat and HSF and SSBs; instead of the eight components reported by Fung et al., [20], who did not include high sugary foods in the “sugars” component but included sodium intake (mg/d), which we did not. All of these components were classified according to a quintile ranking. For those healthy products (fruits, vegetables, nuts and legumes, whole grain, low-fat dairy products) the highest score of 5 was assigned to those in the highest quintile and the lowest score of 1 to those in the lowest quintile, with intermediate values being assigned accordingly. Whereas, for unhealthier products (red and processed meat and solid sugar and SSBs) values were assigned oppositely. A final score was calculated with the sum of the 7 individual values, with a possible value between 7 and 35. The score of seven was for the unhealthiest diet, and 35 for the healthiest.

### 2.6. Covariates

The regular performance of moderate-to-vigorous physical activity was evaluated from two questionnaire questions: 1) “Does your child practice any extracurricular sport?” and 2) “Is your child member of any sports club?” [31]. If either one of the answers was positive, the child was considered as a regularly active child, oppositely, if neither of the answers was positive, the child was considered a non-active child. In addition, information regarding maternal education was obtained from the same questionnaire and categorized in three groups: low (primary school), medium (high school) and high (bachelor´s degree or higher) educational level.

### 2.7. Blood Samples 

Blood samples were collected after a 10–12 hour overnight fast. General biochemical analyses were done at the participating University Hospitals. Serum triglycerides, total cholesterol, low-density lipoprotein cholesterol (LDL-C), high density lipoprotein cholesterol (HDL-C), glucose and insulin were analyzed by automatic microparticle analyzers (Axsym, Abbott Laboratories, Chicago, IL, USA)

Insulin resistance was evaluated by means of the homeostatic model assessment of insulin resistance (HOMA-IR) defined by HOMA-IR= fasting glucose (mmol/L) x fasting insulin (μU/mL)/22.5 [32].

### 2.8. Statistical Analyses

Analyses were performed in the total sample, as well as stratifying by pubertal stage and presence or not of overweight/obesity, as indicated. Normal distribution of the variables was assessed with the Kolmogorov-Smirnov test, and non-normally distributed variables (total cholesterol, serum triacylglycerides, HDL-C, EDSF CF, HSF CF, SSB CF, MDASH score, SBP and DBP) were transformed into a logarithm scale for analyses.

Means and standard deviation were calculated for the studied variables, and student’s t-test was used for simple comparisons between pairs of groups. 

EDSF, HSF, SSB and MDASH score were compared among children with normal BP, risk of elevated BP or risk of hypertension with a general linear model adjusted for sex, age, recruitment center and maternal education. BMI and habitual physical activity (one hour per week or more of extracurricular sport) were also included as co-variables for specific analyses. 

Anthropometry and metabolic variables were compared among EDSF CF groups using general linear models adjusted for sex, age and recruitment center, as well as for BMI, maternal education and physical activity for specific analyses. 

All analyses were carried out with SPSS 21.0.

## 3. Results

First, the general characteristics of the studied population by pubertal and BMI status were included in Appendix A. There was a 19% of children with overweight and a 51.3% of children with obesity among prepubertal children. Whereas in pubertal children we found a 29.3% of children with overweight and a 45.5% with obesity. No differences were found for age and height in pubertal children, regardless of their weight status. Whereas prepubertal children with overweight were older than those with normal weight or obesity. Regarding height, prepubertal children with overweight or obesity were also taller than those with normal weight. The rest of the variables such as BP and biochemical are shown in Appendix A.

Second, the consumption frequency of the studied food groups was compared between the males and females, as well as between prepubertal and pubertal children (Table 1). MDASH scores were significantly healthier in pubertal vs. prepubertal females, and healthier in both prepubertal and pubertal females when compared with their male counterparts.

### 3.1. Blood Pressure and Food Consumption Frequency

Significant differences were found in CF of EDSF between children with normal DBP and diastolic hypertension, whereas no differences were found in the consumption frequency of the different studied food groups among children with normal SBP, risk of elevated SBP or risk of systolic hypertension (Table 2). When analyses were stratified by pubertal stage, this difference disappeared and a new significant difference was found for SSB (*P* = 0.006), that were found to be consumed more often (mean=14.1 times/week, SD = 26.6) by pubertal children with diastolic hypertension Stage I or II than by those with normal DBP (mean= 6.7 times/week, SD = 11.5) (Appendix A). However, no differences were found in prepubertal children (Appendix A). In order to elucidate whether these differences were due to the high obesity prevalence in the sample, stratified analyses by nutritional status were conducted. Normal-weight children showed significant differences among the three diastolic BP levels (Appendix A).

### 3.2. Food Groups Consumption by Quartiles of Frequency of Consumption of Energy Dense Salty Food.

Next, the relationship between the studied food groups was evaluated by comparing the CF of the other food groups among the three EDSF CF groups (below P25th, P25th–P75th and above P75th) (Table 3). Significant differences were found in HSF CF and SSB CF and in the MDASH score among the three EDSF CF groups. HSF and SSB CF being higher for children within the highest quartile of EDSF intake compared with the other groups. Oppositely, the MDASH score was higher for children within the lowest quartile of EDSF. 

When analyses were performed separately by pubertal stage, significant differences were found in HSF CF and MDASH score in prepubertal children, and in SSB CF and MDASH score in pubertal children (Appendix A).

### 3.3. Anthropometric and Metabolic Variables by Quartiles of Frequency of Consumption of Energy Dense Salty Food

Finally, no differences were found among the EDSF CF groups for anthropometric or metabolic variables (Table 4). Once divided by pubertal stage, pubertal children in the highest quartile of EDSF CF (above P75th) showed significantly higher SBP values than those with lower EDSF CF (below P25th or P25th–P75th) when the analysis was adjusted by sex, center, maternal education, BMI and physical activity (Appendix A).

## 4. Discussion

The relationship between salt consumption and BP has been investigated in several studies, but few studies focused on the salt intake from processed foods, which are also rich in saturated fats. This may be due to the former prevalence of single nutrient-based studies. However, nowadays there is a growing interest in the health effects of whole foods and dietary patterns. Following this trend, we investigated the association between the consumption frequency of three food groups defined according to their salt and sugar content, and the DASH dietary pattern, and the presence of systolic or/and diastolic HTN, distinguishing between risk of elevated BP and risk of hypertension (Stages I and II). The most important finding was the association between EDSF CF and diastolic HTN. We also observed that MDASH score was higher in girls compared to boys and in pubertal compared to prepubertal girls. Regarding EDSF CF, we showed its positive association with HSF and SSB dietary patterns and its inverse association with MDASH score.

Regarding the sex and pubertal differences in MDASH score, the obtained results are in line with those observed in the literature. These differences could be explained by a greater concern for physical appearance and weight control in girls. In a longitudinal study did in German, girls showed a decrease in meat intake and increase in whole and refined grains, vegetables, oils and tea intake with puberty onset; however, boys tended to keep a similar diet when reaching puberty [33]. Swedish Females from 17 to 21, also ate fruit and drank tea more often than males [34]. However, there are other determinants that could also influence the dietary intake during puberty such as: family income, urban/rural residence, maternal education and baseline dietary intakes [35].

In the current analyses, we considered not only independent food items or classic food groups, but a combination of foods characterized by high energy and high salt content. We found that children with diastolic hypertension showed higher EDSF CF than those with normal DBP; and no differences were observed for SBP. This could be explained since SBP is more influenced by environmental conditions, such as the measurements during the visit, than DBP, as well as to the fact that it may take more time for SBP than DBP to be altered. To our knowledge, there is no other study in children considering the relationship between BP and EDSF CF. One Italian study found high sodium snacks to be positively associated with SBP and DBP in adolescents [36]. However, Julián-Almarcegui et al., showed that snack consumption was associated with increased SBP and MAP, and not with DBP [28]. The National Diet and Nutrition Survey for young people from the UK found that an increase of 1g/day in sodium intake was associated with an increase of 0.4 mmHg in SBP [9]. In line with our findings, a study with 1440 participants from Iran and mean age 12.42 years found a positive association between “junk food” (salty snacks, sweets, sugar-sweetened beverages and fast foods) and SBP and DBP [37]. However, in the present study, salty foods, sugar foods and beverages were considered separately in EDSF, HSF and SSB, respectively.

Pubertal children with diastolic hypertension I or II had higher SSB CF than those with normal DBP. Similarly, a study showed an association between SSB intake and DBP in children of 11–12 years old but not at 5–6 years; also finding an increase in SSB consumption with age [16]. Other studies one from Tehran and the other from China respectively found an association between SSB intake and hypertension in children with any kind of hypertension [38] and with systolic hypertension [39]. While another study in Iranian children 6–18 years old did not find an association [40]. Moreover, a survey in the US showed that adolescents and older children consumed more SSBs than younger children [41]. These previous studies suggest that the strength of the association between BP and SSB could increase with growth. A proposed hypothesis for this association is the activation of the sympathetic nervous system and, consequently, a rise in SBP, by SSB intake [16]. However, the design of the present study did not allow to confirm this hypothesis.

We found no effect of MDASH score on SBP nor DBP, despite many studies have shown its association with BP, mainly in adults. In Iranian children, DASH score measured with a semiquantitative FFQ was associated with lower SBP values [22,42]. Other study in US youth showed an inverse association between DASH score and DBP, but DASH was measured by 24h recalls and with nine target nutrients [43]. A longitudinal study found a reduction in SBP levels in American children (3 to 5.9 years) with a higher DASH score, in this case DASH was measured by a three days dietary records [44]. Couch et al., [23] found a significant decrease of SBP in adolescents with a clinical diagnosis of prehypertension or hypertension (11 to 18 years) who followed a DASH diet for three months. In contrast, Saneei et al., [45] found that DASH diet prevented the rise of DBP but had no effects on SBP. Our results are not in line with these previous studies, since we found no association between this score and SBP or DBP. However, ours is an observational study in which MDASH score was obtained by a qualitative FFQ, which hinders its comparison with other studies.

Regarding the effect of BMI in the association between EDSF CF and BP, in the stratified analyses differences were only observed in children with normal weight, although numbers were small in the pre-hypertensive and hypertensive groups. In contrast, Yang et al., showed a stronger association between BP values and salt intake in children with obesity or overweight status [12].

Following this, the relationship between the EDSF CF and the other food groups was studied. We observed strong and direct associations between EDSF and HSF and SSB CFs and an inverse association between EDSF CF and MDASH score. To explain the positive associations, it could be argued that ESDF consumption causes thirst and increases fluid consumption in the form of SSB. In turn, the higher SSB consumption would increase insulin secretion enhancing appetite and leading to an increase in food consumption, specially processed food [46]. Interestingly, after stratifying by pubertal stage, the association between EDSF CF and HSF CF remained only in prepubertal children, the association between EDSF CF and SSBs remained only in pubertal children and the inverse association between EDSF CF and MDASH score remained significant in both pubertal stages. The association between EDSF CF an SSB CF in older children agrees with many previous studies [47]. However, there is a lack of studies evaluating the association between sugary solid foods and other dietary patterns, but we should not forget high sugary solid foods, given their frequent consumption by prepubertal children. Our results support both, solid and liquid sugary foods must be taken into account when designing nutrition polices. In accordance with our results, aiming for the maintenance of DBP below the 90th percentile, we suggest a reduction for EDSF of less than 10 a week. As for SSB and HSF we suggest limiting their consumption according to individual country guidelines. However, we suggest promoting their exchange in favor of a higher consumption of healthy foods, since this practice has been more effective for the general population [48].

Finally, to our knowledge this is the first study to analyze the effect of EDSF CF on anthropometric and metabolic variables. In contrast with other studies, no differences were found regarding metabolic variables. However, those previous studies considered fried snacks and candies [49], or western dietary patterns [50,51,52,53,54]. As for anthropometry, no differences were observed between the different EDSF CF groups. These results agree with others who have not found significant differences in BMI and WC between children who followed a “western” dietary pattern or not [55,56,57]. However, other studies have shown significant differences in BMI and WC between an “unhealthy” or a “healthy” dietary pattern [52,58]. These discrepancies could be due to the heterogeneous origin and methodology of the studies, since there was only one European study, more specifically a Spanish study, and since each study used different dietary survey. 

In light of our findings, the pubertal stage appears as an important factor to be considered in future studies. We report several differences in the association between hypertension and different food groups CF between prepubertal and pubertal children. These could be due to the previously mentioned relationship between diet and puberty, as well as to the hormonal changes that occur which can influence lipid levels, body fat and lean mass distribution [59]. 

One of our limitations was that we were unable to quantify salt and sugar intake because we used a qualitative FFQ without standard serving sizes derived from another study with different main objectives and, thus, not designed to evaluate sodium intake in children and adolescents with risk of hypertension. In addition, we could not use the "gold standard" 24-hour urinary excretion for the assessment of sodium intake. As for the use of an oscillometric device to measure blood pressure instead of auscultation or ambulatory blood pressure monitoring, this was due to its feasibility and in order to increase compliance, since these devices do not need prior preparation and are valid for an initial BP measurement. In addition, their measurements are reproducible and feasible in other contexts outside the hospital environment such as schools.

In contrast, a positive aspect was the good sample size that covered a wide age range, as well as the broad number of clinical and anthropometric variables. Indeed, the classification by Tanner stage allowed us to find differences between the developmental stages in the association of EDSF CF with hypertension. Regarding the foods included in EDSF group, we focused on processed food sources, also high in saturated fats, to classify foods. Another source of salt is bread and same bakery products. We did not included bread for different reasons; first, Spanish bakers and the Spanish food safety and nutrition agency (AECOSAN, health ministry) have reached an agreement to reduce bread salt content. For this reason, although bread is a salt source, we did not include it into the EDSF group. Therefore, we decided to include foods with a sodium content above 500 mg / 100 gr and salt bread content is around 490 and 520 depending on the source [60,61]. Finally, bread consumption is very common across Spanish children, which could mask the potential more subtle differences due to the consumption frequency of other food products. 

Our results show that sex and pubertal stage influence MDASH score, and children with diastolic hypertension showed a higher EDSF CF. In addition, pubertal children with diastolic hypertension showed a higher SSB CF. In addition, EDSF CF was positively associated with HSF and SSB CFs and inversely associated with MDASH score. SBP was found to be increased in the highest EDSF CF quartile in pubertal children. No association was found for the rest of metabolic and anthropometric variables among EDSF CF groups. Our findings support the existing recommendations limiting the consumption of sugar-sweetened foods and beverages, as well as processed EDSF, and support the promotion of the DASH diet, in order to prevent HTN in children and adolescents. The present results highlight the importance of dietary tracking during childhood, especially in prepubertal children. Policies and intervention programs must be established that aim to reduce salt consumption, not only in the form of table salt, but mainly as added salt in snacks and processed foods. Moreover, our results suggest that those policies and programs would lead to a reduction in HSF and SSB consumption as well. 

## Figures and Tables

**Table 1 nutrients-12-01027-t001:** Food group consumption according to sex and pubertal stage.

	Male						Female					
	Prepubertal			Pubertal			Prepubertal			Pubertal		
	N	Mean	SD	N	Mean	SD	N	Mean	SD	N	Mean	SD
EDSF	180	13.1^a^	8.8	138	12.7^a^	7.8	157	13^a^	9.0	207	12^a^	8.9
HSF	180	20.8^a^	11	138	20.6^a^	13.7	157	21.2^a^	13.7	207	20^a^	12.3
SSB	180	8.8^a^	16.8	138	8.1^a^	13	157	9.2^a^	15.3	207	7.5^a^	15.1
MDASH	180	17.3^a^	4.7	138	17.8^a^	5.3	157	18.5^a*^	5.1	207	19.4^b*^	4.6

Different superscript letters indicate significant differences (*p* < 0.05) in the t-student test between prepubertal and pubertal children and an asterisk means significant differences (*p* < 0.05) for comparisons between males and females of the same pre/pubertal stage. EDSF: energy dense salty food (times/week); HSF: high sugar food (times/week); MDASH: modified dietary approach stop hypertension (score); SD: standard deviation; SSB: sweet sugar beverages (times/week).

**Table 2 nutrients-12-01027-t002:** Food group consumption according to blood pressure stage.

	Normal BP				Risk of elevated BP				Risk of Hypertension Stage I and II				Linear General Model *p*-Value		
	N	Mean	SD	95%CI	N	Mean	SD	95%CI	N	Mean	SD	95%CI	P1	P2	P3
**SBP**	453	102^a^	11	(102 -104)	57	117^b^	5	(114-119)	124	125^c^	9	(123-127)	**0.000**	**0.000**	**0.000**
EDSF	441	12.74	8.55	(11.96-13.51)	55	13.91	8.91	(12.05-16.47)	120	12.39	8.11	(10.71-13.80)	0.351	0.290	0.316
HSF	346	20.88	12.67	(19.12-21.70)	42	22.66	14.59	(18.79-26.28)	93	19.27	10.19	(18.02-23.19)	0.354	0.250	0.551
SSB	436	7.58	14.44	(6.41-9.20)	55	10.10	13.87	(6.40-14.30)	118	8.75	15.80	(5.33-10.87)	0.599	0.626	0.512
MDASH	441	18.06	4.91	(17.83-18.70)	55	18.04	4.82	(16.33-18.82)	120	18.85	5.00	(17.33-19.07)	0.283	0.468	0.657
**DBP**	532	62^a^	7	(61-63)	28	74^b^	3	(72-77)	74	84^c^	9	(82-86)	**0.000**	**0.000**	**0.000**
EDSF	517	12.4^a^	7.9	(11.76-13.15)	28	11.9^a,b^	7.1	(8.96-14.96)	71	15.6^b^	12.3	(13.55-17.33)	**0.013**	**0.005**	**0.008**
HSF	404	20.9	12.5	(19.65-22.05)	21	19.9	15.1	(14.60-25.14)	57	19.9	11.0	(16.86-23.27)	0.858	0.993	0.650
SSB	511	7.8	14.0	(6.58-9.09)	28	4.9	9.3	(0.09-10.81)	71	10.7	19.7	(7.11-13.86)	0.221	0.227	0.226
MDASH	517	18.1	4.9	(17.77-18.58)	28	18.4	5.0	(16.46-19.95)	71	18.6	5.4	(17.41-19.61)	0.851	0.776	0.890

Different superscript letters (a, b) indicate significant differences (*p* < 0.05) among normal BP, risk of elevated BP and risk of hypertension I or II. The general linear model was adjusted for center, sex, age, maternal education (P1), plus body mass index (BMI) (P2), plus physical activity (P3). CI: confidence interval; EDSF: energy dense salty food (times/week); HSF: high sugar food (times/week); MDASH: Modified Dietary Approach Stop Hypertension (score); SD: standard deviation; SSB: sweet sugar beverages (times/week).

**Table 3 nutrients-12-01027-t003:** Food group consumption by quartiles of frequency of consumption of energy dense salty food.

	<P25th EDSF				P25th-P75th EDSF				>P75th EDSF				Linear General Model *p*-Value		
	N	Mean	SD	95%CI	N	Mean	SD	95%CI	N	Mean	SD	95%CI	P1	P2	P3
HSF	168	17.0^a^	11.4	(16.01–20.73)	347	20.1^a^	12.1	(18.64–21.66)	168	24.2^b^	13.4	(21.07–25.09)	0.004	0.005	0.010
SSB	168	5.9^a^	12.1	(2.22–7.0)	347	7.9^a^	15.1	(5.54–8.64)	168	11.8^b^	17.5	(10.87–15.48)	0.000	0.000	0.000
MDASH	168	20.2^a^	5.0	(18.64–20.15)	347	18.2^b^	4.7	(17.52–18.51)	168	16.6^b^	4.8	(16.57–8.03)	0.000	0.000	0.001

Different superscript letters (a, b) indicate statistical significance in the general linear model among the quartiles of frequency of consumption of energy dense salty foods, adjusted for center, sex, age, maternal education (P1), plus BMI (P2), plus physical activity (P3). CI: confidence interval; EDSF: energy dense salty food; HSF: high-sugary foods (times/week); MDASH: Modified Dietary Approach to Stop Hypertension (score); SD: standard deviation; SSB: sugar-sweetened beverages (times/week).

**Table 4 nutrients-12-01027-t004:** Anthropometric and metabolic variables by quartiles of frequency of consumption of energy dense salty food.

	<P25th EDSF			P25th-P75th EDSF			>P75th EDSF			Linear General Model *p*-Value		
	N	Mean	SD	N	Mean	SD	N	Mean	SD	P1	P2	P3
Age	168	11	2.7	347	10.5	2.4	168	10.6	2.3	0.289	0.491	0.869
Weight (kg)	168	54.6	18.0	347	51.4	19.6	168	51.3	19.9	0.985	0.340	0.600
WC (cm)	168	82.4	13.8	347	78.6	16.7	168	78.1	16.7	0.793	0.391	0.519
MAP (mmHg)	168	79.8	10.4	347	79.07	9.39	168	80.69	9.47	0.180	0.125	0.150
SBP (mmHg)	168	109	14	347	108	14	168	109	14	0.348	0.290	0.335
DBP (mmHg)	168	65	11	347	65	10	168	66	10	0.252	0.208	0.228
TAG (mg/dL)	168	67	31	347	70	35	168	70	31	0.522	0.401	0.424
CHOL (mg/dL)	168	164	34	347	162	26	168	164	30	0.563	0.552	0.544
LDL-C (mg/dL)	168	96	31	347	93	24	168	94	24	0.749	0.766	0.775
HDL-C (mg/dL)	168	50	12	347	52	15	168	55	15	0.309	0.295	0.359
Glucose (mg/dL)	168	84	9	347	85	8	168	86	7	0.955	0.972	0.981
Insulin (mU/l)	168	11.7	7.3	347	12.4	8.7	168	12.2	11.3	0.528	0.228	0.546
HOMA-IR	168	2.50	1.65	347	2.62	1.90	158	2.61	2.46	0.706	0.374	0.727

General linear model adjusted for center, sex, age, maternal education (P1), plus BMI (P2), plus physical activity (P3). CHOL: total cholesterol; DBP: diastolic blood pressure; HDL-C: high density lipoprotein cholesterol; HOMA-IR: homeostasis model assessment for insulin resistance; LDL-C: low density lipoprotein cholesterol; MAP: mean arterial blood pressure (2*DBP+SBP)/3; SBP: systolic blood pressure; SD: standard deviation; TAG: triglycerides; WC: waist circumference.

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
