# Peer review of "Energy Dense Salty Food Consumption Frequency Is Associated with Diastolic Hypertension in Spanish Children"

_nutrients, 2020, doi:10.3390/nu12041027_

Round 1
Reviewer 1 Report
This study aimed to investigate whether children with risk of elevated BP had a higher consumption frequency (CF) of energy-dense salty foods (EDSF), high-sugary foods (HSF) and sugar-sweetened beverages (SSB) or a low DASH score. The authors found that children with diastolic hypertension showed higher EDSF CF than those with normal DBP, however, no differences were observed for SBP. Moreover, pubertal children with diastolic hypertension I or II had higher SSB CF than those with normal DBP. The results were interesting. However, the following serious concerns should be addressed.
Major comments:
- Previous studies have indicated that high amounts of salt or sugar intake are correlated with the occurrence of hypertension and cardiovascular diseases. in this study, the authors aimed to explore the influence of frequency of high salt or sugar intake on blood pressure regardless of the total amount of daily salt or sugar intake. Please provide sufficient rationale with references to support the importance of analyzing frequency rather than the amount of daily intake of salt and sugar.
- In the abstract, the authors described that diastolic hypertension was associated to higher CF of EDSF in the whole sample and to higher CF of SSB in pubertal children. However, they did not show the results in the manuscript. Please clarify it.
- The demographic data of children, such as age, height, weight, and BMI, need to be added in the results.
- In Table 2, can the authors provide the value of SBP and DBP for the normal BP, risk of elevated BP, and risk of hypertension stage I and II groups? These data would help readers to gain more information regarding how to interpret the results.
- Literature reviews have indicated that SBP is one of the major determinant for the risk of cardiovascular disease and outcomes. In the present study, no significant differences were observed for the association of SBP and higher EDSF CF. Can the authors provide the explanations for this finding?
- In this study, the total amount of high salt or sugar intake was not taken into consideration. The authors need to discuss this influence on the application of results.
- In Table 4, The authors showed that the SBP and DBP of P75th EDSF children were 109 and 66 mmHg. However, these seemed to be within normal range of blood pressure clinically. Can the authors make some explanations or discussion for this part of results?
- In the discussion, the authors need to provide some suggestions about CF of EDSF, HSF and SSB for reducing high blood pressure in children.
Minor comments:
1. The meaning of label of significant differences in each table was not mentioned and clarified, such as the indication of “a”, “ab”, and “b”.
Reviewer 2 Report
Dear Authors,
The work presented is interesting and novel given its new approach to the analysis of salt consumption and its relationship with the consumption of sugar from processed foods.
The work is well justified, the statistical analyses are relevant and well developed and the results are of interest in terms of public health. Nevertheless, I am enclosing the documents with several comments and suggestions for changes, especially in formal aspects or with requests for clarification that I believe will facilitate the reader's understanding of the results and their interpretation.
I hope you will find my comments useful and that they can contribute to enhancing the quality of the work.
Best regards,

Round 2
Reviewer 1 Report
The authors made some revisions according to comments, which improved the manuscript. However, the reference, "Molag ML, De Vries JH, Ocke MC, Dagnelie PC, Van den Brandt PA, Jansen MC, et al. Design Characteristics of Food Frequency Questionnaires in Relation to Their Validity. Am J Epidemiol. 2007; 166(12): 1468-1478" used to support the importance of analyzing frequency rather than the amount of daily intake of salt and sugar, should be integrated into the introduction. After this minor revision, the revised manuscript would be acceptable for publication.
Author Response
Please see the attachment

This manuscript is a resubmission of an earlier submission. The following is a list of the peer review reports and author responses from that submission.
Round 1
Reviewer 1 Report
This study aimed at evaluation of the effect of consumption of energy-dense salty foods (EDSF) and high-sugar foods (HSF) on diastolic hypertension in Spanish children. Based on anthropometry, blood pressure (measured S/D, calculated MAP), and biochemical parameters (lipid profile, glucose and insulin) measured in 687 children, ages 5-16 years, with normal and excess weight (overweight/obese), the authors concluded that diastolic hypertension is associated with consumption frequency of EDSF.
The authors did justice to this clinical study execution and reporting of results. The subject matter is well written and the study performed with due diligence.
Although this study definitely ushers in new refreshing research on hypertension and high sodium intake, please consider revising the discussion section to include current study limitations.
Reviewer 2 Report
This topic is certainly important and interesting, but I have objections to the method of diagnosing hypertension. More appropriate to confirm HTN is 24-hour ambulatory blood pressure monitoring (ABPM) or if elevated BP is suspected on the basis of oscillometric readings,confirmatory measurements should be obtained by auscultation at 3 different visits.
Measurements obtained by oscillometric devices are usually higher compared with readings obtained by auscultation. What about white coat HTN? Is a device for measuring pressure has been validated in the pediatric age group?
Due to the numerous study group I propose topic modification and data presentation depending on age, nutritional status and dietary habits.
Reviewer 3 Report
This is an interesting manuscript addressing a relevant issue, where additional population based data is required. The manuscript is well organized and written in adequate style. However, this article has serious limitations, the main one being that the aim of the study is to investigate the association between elevated blood pressure and consumption of energy-dense salty foods, high sugary foods and sugar-sweetened beverages and the DASH score, but the authors inform dietary intake just reporting bare frequency of consumption, neglecting portion sizes, servings or any attempt to a more accurate diet assessment, useful to achieve the study aim, despite the ample age range of the study sample (5-16 yr.).
Additional comments:
.- Study sample.- The authors recruited the sample for this trial in hospital pediatric nutrition and endocrine departments. Although they report the exclusion criteria, it would be valuable to have some kind of background information about the health problems why those children and adolescents visited those services and clinics.
.- Clinical examination.- The authors report they estimated overweight and obesity in the sample using IOTF (Cole et al.) criteria, but they do not report this information in this article. In addition, they mention they used Spanish reference standards to calculate BMI Z-scores. To do so, they used the reference standards issued in 2004, Why did they not use the most recent standards issued in 2011? In addition, Did the authors use BMI or BMI z-scores for adjustment in the general lineal models? This should be clarified.
.- Dietary intake.- This section is the most problematic in this article, as already commented on.
Some additional issues in this regard:
.- The focus of this article was the assessment of energy-dense salty food, high-sugary foods and sugar sweetened beverages in this sample of Spanish children and adolescents. Published data on dietary sources of sodium intake in Spanish children and adolescents, using 24 hour urine sodium as a biomarker, and additional studies, report bread, salted nuts and olives as important sources of dietary sodium, but those are not considered in this study.
In line with the above, regarding sugary foods and sugar sweetened beverages, what included the term "cakes" in the FFQ? Why did not include sugary processed dairy desserts? and regarding sugar-sweetened beverages, why the author did not consider sugar added flavored milks, milk shakes and drinking yogurt?
.- In the methods section, the authors report they calculated the DASH score. The DASH score, as reported by Fung, is computed based on more accurate diet assessment, adjusted for energy intake. In addition, the authors did not estimate sodium intake but consider high sugary foods. Therefore, it would be more appropriate to refer to a modified DASH score or a DASH-like score, rather than DASH score.
.- Results.-
Given the standard deviation in the frequency of consumption, it would be informative to see the confidence intervals of the difference between groups.
The heading 3.2 "Relationship between food groups consumption by the different energy-dense salty food consumption", the text under that heading and the title of table 3 would be more understandable as food group consumption by quartiles of frequency of consumption of energy dense salty food.
The same for heading 3.3 and title of table 4: Anthropometric and metabolic variables by quartiles of frequency of consumption of energy dense salty food.
Discussion
It would be valuable to refer to previous published data on dietary sources of salt and added sugars in the diets of Spanish children and adolescents.
The authors should comment on the limitations of this study.
The quality and value of this manuscript will considerably benefit if the authors could reanalyze the food frequency data considering serving sizes and run again data analysis.